

# Fusarium head blight in the Russian Far East: 140 years after description of the 'drunken bread' problem

Tatiana Gagkaeva, Aleksandra Orina and Olga Gavrilova

Laboratory of Mycology and Phytopathology, All-Russian Institute of Plant Protection, St. Petersburg, Pushkin, Russian Federation

## ABSTRACT

The first appearance of Fusarium head blight (FHB)—and the beginning of scientific research of this disease—occurred the Far East region of Russia at the end of the 19th century. In the summer of 2019, in the Amur region, which comprises 60–70% of grain production in the Russian Far East, flooding caused a state of emergency. The quality of wheat and barley grains grown under natural conditions of FHB outbreaks, including grain infection, fungal species composition, DNA content of *F. graminearum* and chemotypes, and the presence of various mycotoxins, was studied. *Fusarium* infection rates reached extremely high percentages, 51–98%, the majority of which were *F. graminearum* infections. The amount of *F. graminearum* DNA in wheat grain samples was higher than in the barley grain samples and averaged 6.1 and 2.1 pg/ng, respectively. The content of deoxynivalenol (DON) in the wheat samples reached 13,343 ppb and in barley reached 7,755 ppb. A multilocus genotyping assay was conducted on the partially sequenced fragments of the translation elongation factor EF-1a, ammonium ligase gene, reductase gene, and 3-O-acetyltransferase gene in 29 *Fusarium graminearum sensu lato* strains from the grain harvested in the Amur region. All strains from the Far East region were characterized as *F. graminearum sensu stricto*; 70% were the 15-AcDON chemotype, while the other strains were the 3-AcDON chemotype. According to the results, after 140 years of study of FHB, we are still not very successful in controlling this disease if conditions are favorable for pathogen development. Even at present, some of the grain harvested must be destroyed, as high contamination of mycotoxins renders it unusable.

## INTRODUCTION

The first description of Fusarium head blight (FHB) within the territory of Russia was in the Far East in 1882 (*Palchevsky, 1891*; *Voronin, 1890*). This region is typically a monsoon climate with very damp and warm summers due to the influences of the Sea of Japan and the Pacific Ocean. The scientific investigation of this disease began at the end of the 19th century, but long before this, Chinese peasants and later Russian settlers related the poisoning of people and animals with pinkish grains and heads in the fields.

Corresponding author
Tatiana Gagkaeva,
t.gagkaeva@yahoo.com

Between 1882 and 1914, epidemics of this disease in the Far East occurred almost every year (*Naumov, 1916*). Consumption of affected grain and straw caused numerous cases of food poisoning of people and farm animals. The initial signs and symptoms of the disease resemble those that can develop after drinking too much alcohol (including dizziness and headache, trembling hands, confusion, and vomiting) and thus the disease was named 'drunken bread'. The extensive research undertaken by Russian mycologists revealed that *Fusarium roseum* Link (*F. graminearum* Schwabe) with teleomorph stage *Gibberella saubinetii* Sacc. (*G. zeae* [Schwein.] Petch) was the principal cause of the disease (*Jaczewski, 1904*; *Naumov, 1916*; *Voronin, 1890*).

*Palchevsky (1891)*, who lived in this territory and was one of the first to report the disease of grain crops, studied its etiology and deposited diseased grain head specimens in herbaria (the first specimens, kept in the Herbarium LEP of our laboratory, are dated 1912). Thanks to this inquisitive individual, drawings of typical symptoms of the disease and pathogens were published (Fig. 1).

FHB was a persistent problem in the Far East during the 20th century (*Abramov, 1938*; *Naumov, 1916*) and continues to be today. High severities of FHB are reported nearly every year in the region. Mycological analyses of seed samples from 1998–2002 have shown a high level of FHB-infected wheat and barley seed (23–32%). The most frequently isolated pathogen was *F. graminearum* (*Gagkaeva et al., 2002*; *Ivashchenko, Shipilova & Levitin, 2000*).

Potential toxic effects of mycotoxins associated with FHB, particularly trichothecenes, which are secondary metabolites produced by *F. graminearum* and other *Fusarium* species, , can result in numerous health problems after consumption of infected grain, flour, and processed products. *Fusarium* outbreaks are a concern because of loss of grain yield and quality and mycotoxin contamination.

The development of multilocus sequence typing (MLST) has facilitated the identification of species and chemotypes of the *F. graminearum* species (*Fg*) group (*Ward et al., 2008*). Among them, the ubiquitous *F. graminearum sensu lato* (*s. lat.*) includes at least 16 phylogenetic species (*Aoki et al., 2012*; *O'Donnell et al., 2000*; *O'Donnell et al., 2004*; *O'Donnell et al., 2008*) united into the *Fg* group. Based on MLST assays, several species of the *Fg* group, including *F. graminearum sensu stricto* (s. str.), *F. ussurianum* T. Aoki, Gagkaeva, Yli-Mattila, Kistler & O'Donnell, and *F. vorosii* B. Tóth, Varga, Starkey, O'Donnell, H. Suga & T. Aoki, were identified in the grain grown in the Russian Far East (*Yli-Mattila et al., 2009*). A biogeographic hypothesis suggests that *F. vorosii*, *F. ussurianum*, and *F. asiaticum* O'Donnell, T. Aoki, Kistler & Geiser may be endemic Asian species within the *Fg* group (*O'Donnell et al., 2004*).

All species within the *Fg* group are capable of producing type B trichothecenes, but the activity of their formation is largely different. Three types of chemotypes have been identified among the strains: deoxynivalenol (DON) and 3-acetyldeoxynivalenol (3-AcDON), DON and 15-acetyldeoxynivalenol (15-AcDON), and nivalenol and 4-acetyl-nivalenol (NIV) (*Moss & Thrane, 2004*; *Ward et al., 2002*).

High humidity and heavy rainfall stimulate the development of *F. graminearum s. lat.* in grain and, as a result, increase its contamination by DON (*Aldred & Magan, 2004*; *Ramirez*,

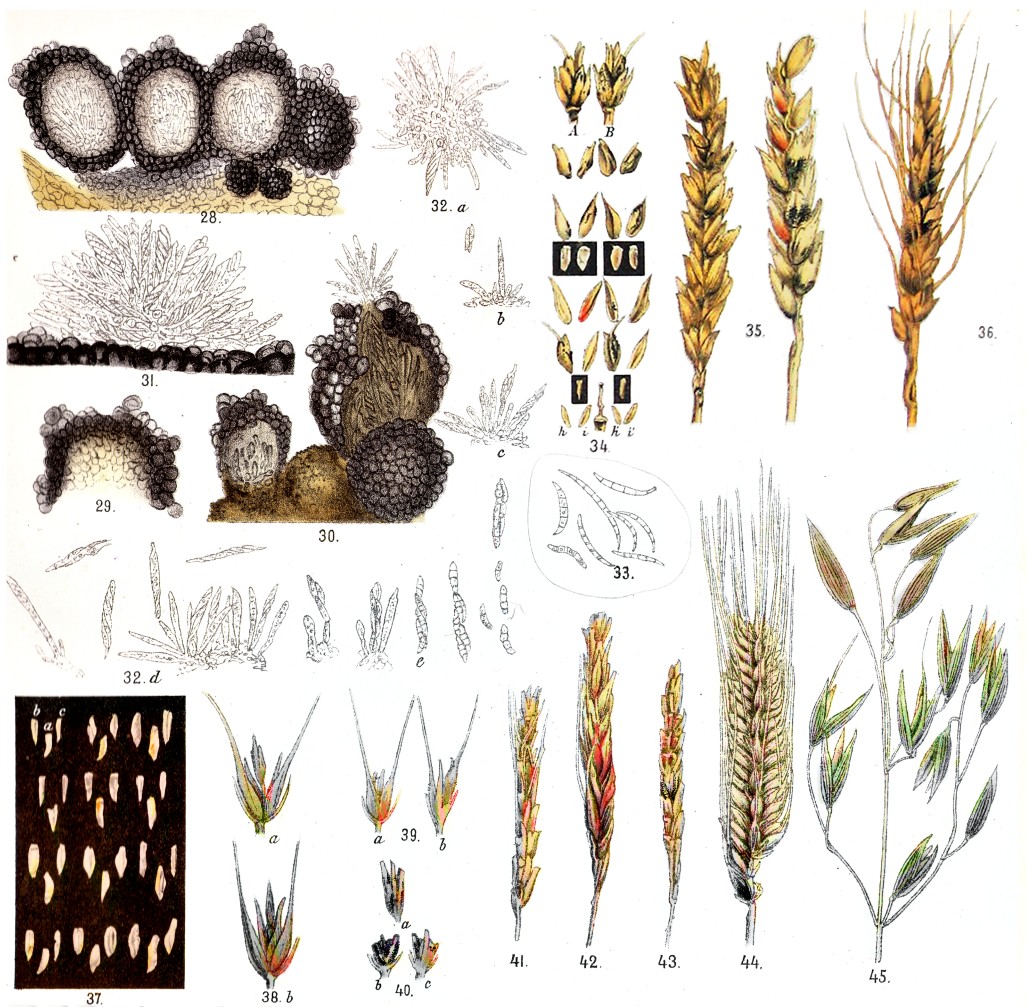

**Figure 1** The fungal perithecia and spores, and the symptoms of *Fusarium* disease of cereals from the Far East of Russia presented in the book by *Palchevsky (1891)*.

*Chulze & Magan, 2006*). The Russian Far East is often exposed to weather disasters, such as floods, which can lead to negative consequences insurmountable by human efforts and technologies resulting in significant agricultural damage. In the summer of 2019, in the Amur region, which accounts for 60–70% of grain production in the Russian Far East, flooding after substantial rainfall has wreaks havoc with extensive damaged crops. In this region, an emergency regime was established on July 25, 2019, and 'about 250,000 ha was flooded, which amounted to about 20% of the total cultivated area in the region' (*TASS, 2019*). As a result, the yield of cereals was only partially saved and harvested.

Despite the long history of the problem in the Far East, there is still no objective information on infection and mycotoxin contamination of harvested grain. Epidemics of FHB in the region fuelled our interest in investigating this disease using available modern methods of research.

**Table 1** Climatic data during the growing season of 2019 in the Amur region ( https://rp5.ru/)

| Month | Average temperature, °C | | | Average humidity, % | Total rainfall, mm | Days with precipitation |
|---|---|---|---|---|---|---|
| | mean | min | max | | | |
| May | +12.0 | −1.6 | +27.5 | 56 | 17 | 19 |
| June | +18.2 | +7.2 | +30.1 | 66 | 46 | 21 |
| July | +21.1 | +13.5 | +30.2 | 83 | 300 | 27 |
| August | +18.6 | +10.6 | +29.2 | 82 | 206 | 25 |

Broad geographic surveys of *Fusarium* species on cereal crops are important to establish if any present shifts in populations occur in response to environmental change. We expected the diversity of *Fusarium* species belonging to the *Fg* group in this territory to be high, since the conditions were very favorable for fungi and led to the disease epidemic. In addition, we assumed that *F. asiaticum* may appear in the complex of pathogens, since in the neighbouring countries of China and Japan this species is detected on cereals with a high frequency (*Gale et al., 2002*; *Láday et al., 2004*; *Qu et al., 2007*; *Suga et al., 2008*). In China, where the problem of FHB is also acute, two species of the *Fg* group have been identified: *F. graminearum* s. str. and *F. asiaticum* (*Qu et al., 2007*). In the north of China, closest to the Amur Region, mostly *F. graminearum* was dominant, and all of strains were the 15-AcDON chemotypes (*Shen et al., 2012*). But *F. asiaticum* was the predominant species in the Yangtze River Basin, and chemotypes of strains were either 3-AcDON or NIV, with 3-AcDON being predominant.

This study aimed was to investigate (1) the natural *Fusarium* species occurrence and mycotoxin contamination of grain from the Amur region in the Far East in the most favorable conditions for pathogens and (2) provide the multilocus analysis of isolated strains of *F. graminearum* sensu lato to species and trichothecene genotype diversity.

## MATERIALS & METHODS

### Grain samples and climatic conditions of growth

In mid-August 2019, grain harvest samples were collected in various flood-rescued fields located in the Amur Region, the Russian Far East. These samples were spring wheat (nine samples of the most common Aryuna variety) and barley (four samples of the most common Acha variety).

Collecting the representative samples from harvesting at these locations was approved by the Russian Science Foundation (project number: 19-76-30005).

The weather in the summer period of 2019 was characterized by disastrous excessive moisture: the total rainfall in July and August was 2.2 and 1.7 times greater, respectively, than the average means of long-term observations (according to https://rp5.ru/). In addition, the number of days with precipitation in these months was 50% and 39% more, respectively, than the average means of the climatic norm (Table 1).

## Mycological analysis of grain

Microscopic examination was conducted to reveal the presence of infected grains and perithecia on seed surfaces, and photographs were taken under Olympus BX53 and Olympus SZX16 microscopes.

One hundred seeds per sample were chosen at random and surface disinfected by soaking in a 5% sodium hypochlorite solution for 3–4 min. Then the grains were washed with sterile water and put into Petri dishes on potato sucrose agar medium (PSA) containing 1 mL/L of an antibiotics solution (HyClone™, Austria). Moreover, a commonly used detergent Triton X-100 (Panreac, Spain) which reduces the linear fungal growth (0.4 μL/L) was added. After 7–14 days of incubation in the dark at 24 °C, identification and demarcation of taxa were carried out (*Gerlach & Nirenberg, 1982*; *Leslie & Summerell, 2006*). The grain infection by the specific taxon of fungi was calculated as the ratio of the number of grains from which these fungi were isolated to the total number of analyzed grains and expressed as the incidence percentage.

## DNA extraction and quantification

The grain samples (20 g) were homogenized separately using sterilized grinding chambers of a batch mill Tube Mill Control (IKA, Königswinter, Germany). The grain flour was stored at −20 °C.

The total DNA from 200 mg of grain flour was isolated using the Genomic DNA Purification Kit (Thermo Fisher Scientific, Vilnius, Lithuania) according to the manufacturer's protocol and as previously described in *Gagkaeva et al. (2019)*. Using the same kit, DNA was also isolated from the mycelium of *Fusarium* spp. strains cultivated on PSA. DNA concentrations from the grain samples and fungal strains were determined using a Qubit 2.0 Fluorometer with a Quant-iT dsDNA HS Assay Kit (Thermo Fisher Scientific, Waltham, MA, USA). Before the start of quantitative PCR (qPCR), the concentrations of all DNA samples were normalized to 23–67 ng/μL.

In every total DNA sample extracted from grain flour, the DNA content of the *F. graminearum* and *F. avenaceum* was evaluated by qPCR with TaqMan probes (*Yli-Mattila et al., 2008*). The reaction was carried out in a 20-μL-volume mixture with 10 μL of a 2× TaqM master mix (AlkorBio, St. Petersburg, Russia), 300 nM of each primer, 100 nM of a fluorescent sample (Evrogen, Moscow, Russia), and 2 μL of the corresponding DNA solution.

Additionally, the DNA content of 3-AcDON and 15-AcDON chemotypes of *F. graminearum* was determined using qPCR with SYBR Green (*Nielsen et al., 2012*). All qPCR assays were run using the CFX 96 Real-Time System thermocycler (Bio-Rad, Hercules, CA, USA). All samples were analyzed at least twice.

## Mycotoxin determination by HPLC-MS/MS

The HPLC-MS/MS multi-mycotoxin method was used to detect different fungal secondary metabolites. In the grain samples, 3-AcDON, 15-AcDON, alternariol (AOH), alternariol monomethyl ether (AME), beauvericin (BEA), DON, deoxynivalenol-3-glucoside (DON-3gl), diacetoxyscirpenol (DAS), fumonisins B1, B2, and B3, T-2 toxin, HT-2 toxin, T-2

triol, neosolaniol (NEO), fusarenone X, moniliformin (MON), nivalenol (NIV), tentoxin (TEN), tenuazonic acid (TeA), and zearalenone (ZEN) were analyzed.

The analysis of the mycotoxins was carried out following the described procedure (*Malachová et al., 2014*). Detection and quantification were performed with a QTrap 5500MS/MS system (Applied Biosystems, Foster City, CA, USA) equipped with a TurboV electrospray ionization (ESI) source and a 1,290 series UHPLC system (Agilent Technologies, Waldbronn, Germany). Chromatographic separation was performed at 25 °C on a Gemini® C18-column, 150 × 4.6 mm i.d., with a 5-μm particle size, equipped with a C18 SecurityGuard cartridge, 4 × 3 mm i.d. (all from Phenomenex, Torrance, CA, USA). Elution was carried out in binary gradient mode. Both mobile phases contained 5 mM of ammonium acetate and were composed of methanol/water/acetic acid ratios of 10:89:1 (v/v/v; eluent A) and 97:2:1 (v/v/v; eluent B), respectively. The recovery of mycotoxins from grain ranged from 79% to 105%.

### Genotyping of *Fusarium* spp.

Among isolated fungi that were morphologically assigned to the *Fg* group (nearly 900), 29 monoconidial strains were randomly selected for further molecular analysis. Additionally, five related *Fusarium* strains with various geographic and substrate origins, the taxonomic status of which requires appraisal, were included in the study (Table 2).

To assess the phylogenetic relationships between all the strains tested, fragments of the translation elongation factor EF-1a (*TEF*), ammonium ligase gene (*URA*), reductase gene (*RED*), and 3-O-acetyltransferase gene (*Tri101*) were used. Their amplification was carried out using specific primers EF1/EF2, URA11/URA16, RED1d/RED2, and TRI1013E/TRI1015B, respectively, according to the authors' protocols and instructions (*O'Donnell et al., 2000*; *O'Donnell et al., 2004*; *O'Donnell et al., 2008*).

The sequencing was carried out on an ABI Prism 3500 sequencer (Applied Biosystems, Hitachi, Japan) using the BigDye Terminator v3.1 cycle sequencing kit (Applied Biosystems, USA). To address the phylogenetic relationships among taxa maximum likelihood (ML), maximum parsimony (MP) analysis was conducted using the MEGA X 10.2 program (*Kumar et al., 2018*) as well as Bayesian posterior probability (BP) by MrBayes v. 3.2.1 on the Armadillo 1.1 platform (*Lord et al., 2012*). Nodal support was assessed by bootstrap analysis on 1,000 replicates. Sequence data were deposited in GenBank.

The *Fusarium* spp. a chemotype (3-AcDON, 15-AcDON, or NIV) was determined using PCR with primers Tri13P1/Tri13P2 according to the authors' protocols and instructions (*Wang et al., 2008*).

All tested *Fusarium* strains are maintained in the collection of the Laboratory of Mycology and Phytopathology at the All-Russian Institute of Plant Protection.

### Statistical analysis

Data were analyzed using Microsoft Office Excel 2010 (Microsoft, Redmond, WA, USA) and Statistica 10.0 (StatSoft, Tulsa, OK, USA). The significance of differences between mean values was estimated by Tukey's test (95% confidence level).

**Table 2   *Fusarium* strains included in the study.**

| Species | Strain[a] | Host | Origin | Year | GenBank accession number[b] | | | | Chemotype |
|---|---|---|---|---|---|---|---|---|---|
| | | | | | TEF | URA | RED | Tri101 | |
| *F. acaciae-mearnsii* | NRRL 26752 | acacia | South Africa | | AF212447 | AF212705 | AF212558 | AF212594 | |
| *F. aethiopicum* | NRRL 46710 | wheat | Ethiopia | | FJ240296 | FJ240274 | FJ240252 | FJ240339 | |
| *F. asiaticum* | NRRL 26156 | wheat | China | | AF212452 | AF212710 | AF212563 | AF212599 | |
| *F. austroamericanum* | NRRL 2903 | | Brazil | | AF212438 | AF212696 | AF212549 | AF212585 | |
| *F. boothii* | NRRL 29020 | corn | USA | | AF212443 | AF212701 | AF212554 | AF212590 | |
| *F. cerealis* | NRRL 13721 | potato | Poland | | AF212464 | AF212722 | AF212575 | AF212611 | |
| *F. culmorum* | NRRL 25475 | barley | Denmark | | AF212463 | AF212721 | AF212574 | AF212610 | |
| *F. culmorum* | MFG 58836 | wheat, grain | Russia, Omsk region | 2015 | MW273182[c] | MW273250 | MW273216 | MW892041 | 3-AcDON |
| *F. culmorum* | MFG 59052 | wheat, grain | Russia, Krasnodar region | 2017 | MW273183 | MW273251 | MW273217 | MW892042 | 3-AcDON |
| *F. culmorum* | MFG 60755 | barley, grain | Russia, Tyumen region | 2015 | MW273187 | MW273255 | MW273221 | MW892043 | 3-AcDON |
| *F. gerlachii* | NRRL 36905 | wheat | USA | | DQ459742 | DQ459776 | DQ459793 | DQ452409 | |
| *F. graminearum* | NRRL 5883 | corn | USA | | AF212455 | AF212713 | AF212566 | AF212602 | |
| *F. graminearum* | MFG 60765 | wheat, grain | Russia, Amur region | 2019 | MW273157 | MW273225 | MW273191 | MW273259 | 15-AcDON |
| *F. graminearum* | MFG 60766 | wheat, grain | Russia, Amur region | 2019 | MW273168 | MW273236 | MW273202 | MW273270 | 15-AcDON |
| *F. graminearum* | MFG 60767 | wheat, grain | Russia, Amur region | 2019 | MW273176 | MW273244 | MW273210 | MW273278 | 15-AcDON |
| *F. graminearum* | MFG 60768 | wheat, grain | Russia, Amur region | 2019 | MW273177 | MW273245 | MW273211 | MW273279 | 15-AcDON |
| *F. graminearum* | MFG 60769 | wheat, grain | Russia, Amur region | 2019 | MW273181 | MW273249 | MW273215 | MW273283 | 15-AcDON |
| *F. graminearum* | MFG 60770 | wheat, grain | Russia, Amur region | 2019 | MW273186 | MW273254 | MW273220 | MW273286 | 3-AcDON |
| *F. graminearum* | MFG 60771 | wheat, grain | Russia, Amur region | 2019 | MW273188 | MW273256 | MW273222 | MW273287 | 15-AcDON |
| *F. graminearum* | MFG 60772 | wheat, grain | Russia, Amur region | 2019 | MW273189 | MW273257 | MW273223 | MW273288 | 3-AcDON |
| *F. graminearum* | MFG 60773 | wheat, grain | Russia, Amur region | 2019 | MW273190 | MW273258 | MW273224 | MW273289 | 15-AcDON |
| *F. graminearum* | MFG 60774 | wheat, grain | Russia, Amur region | 2019 | MW273158 | MW273226 | MW273192 | MW273260 | 15-AcDON |
| *F. graminearum* | MFG 60775 | wheat, grain | Russia, Amur region | 2019 | MW273159 | MW273227 | MW273193 | MW273261 | 15-AcDON |

**Table 2** (*continued*)

| Species | Strain[a] | Host | Origin | Year | GenBank accession number[b] | | | | Chemotype |
|---------|--------|------|--------|------|-----|-----|-----|--------|-----------|
| | | | | | *TEF* | *URA* | *RED* | *Tri101* | |
| *F. graminearum* | MFG 60776 | wheat, grain | Russia, Amur region | 2019 | MW273160 | MW273228 | MW273194 | MW273262 | 15-AcDON |
| *F. graminearum* | MFG 60777 | wheat, grain | Russia, Amur region | 2019 | MW273161 | MW273229 | MW273195 | MW273263 | 15-AcDON |
| *F. graminearum* | MFG 60778 | wheat, grain | Russia, Amur region | 2019 | MW273162 | MW273230 | MW273196 | MW273264 | 3-AcDON |
| *F. graminearum* | MFG 60779 | wheat, grain | Russia, Amur region | 2019 | MW273163 | MW273231 | MW273197 | MW273265 | 15-AcDON |
| *F. graminearum* | MFG 60780 | wheat, grain | Russia, Amur region | 2019 | MW273164 | MW273232 | MW273198 | MW273266 | 3-AcDON |
| *F. graminearum* | MFG 60781 | barley, grain | Russia, Amur region | 2019 | MW273165 | MW273233 | MW273199 | MW273267 | 15-AcDON |
| *F. graminearum* | MFG 60782 | barley, grain | Russia, Amur region | 2019 | MW273166 | MW273234 | MW273200 | MW273268 | 3-AcDON |
| *F. graminearum* | MFG 60783 | barley, grain | Russia, Amur region | 2019 | MW273167 | MW273235 | MW273201 | MW273269 | 3-AcDON |
| *F. graminearum* | MFG 60784 | barley, grain | Russia, Amur region | 2019 | MW273169 | MW273237 | MW273203 | MW273271 | 3-AcDON |
| *F. graminearum* | MFG 60785 | barley, grain | Russia, Amur region | 2019 | MW273170 | MW273238 | MW273204 | MW273272 | 3-AcDON |
| *F. graminearum* | MFG 60786 | barley, grain | Russia, Amur region | 2019 | MW273171 | MW273239 | MW273205 | MW273273 | 15-AcDON |
| *F. graminearum* | MFG 60787 | barley, grain | Russia, Amur region | 2019 | MW273172 | MW273240 | MW273206 | MW273274 | 15-AcDON |
| *F. graminearum* | MFG 60788 | barley, grain | Russia, Amur region | 2019 | MW273173 | MW273241 | MW273207 | MW273275 | 15-AcDON |
| *F. graminearum* | MFG 60789 | wheat, grain | Russia, Amur region | 2019 | MW273174 | MW273242 | MW273208 | MW273276 | 15-AcDON |
| *F. graminearum* | MFG 60603 | barley, grain | Russia, Amur region | 2019 | MW273175 | MW273243 | MW273209 | MW273277 | 15-AcDON |
| *F. graminearum* | MFG 60612 | wheat, grain | Russia, Kemerovo region | 2019 | MW273178 | MW273246 | MW273212 | MW273280 | 3-AcDON |
| *F. graminearum* | MFG 60610 | wheat, grain | Russia, Amur region | 2019 | MW273179 | MW273247 | MW273213 | MW273281 | 15-AcDON |
| *F. graminearum* | MFG 60611 | wheat, grain | Russia, Amur region | 2019 | MW273180 | MW273248 | MW273214 | MW273282 | 15-AcDON |
| *F. graminearum* | MFG 60706 | soybean, leaves | Russia, Amur region | 2019 | MW273185 | MW273253 | MW273219 | MW273285 | 15-AcDON |
| *F. pseudograminearum* | NRRL 28334 | *Medicago* sp. | South Africa | | AF212470 | AF212729 | AF212580 | AF212617 | |
| *F. ussurianum* | NRRL 45665 | wheat, grain | Russia, Jewish autonomous region | 2002 | FJ240300 | FJ240279 | FJ240257 | FJ240344 | |
| *F. vorosii* | NRRL 45790 | wheat, grain | Russia, Primorsky Krai | 2006 | FJ240302 | FJ240281 | FJ240259 | FJ240346 | |

**Table 2** (*continued*)

| Species | Strain[a] | Host | Origin | Year | GenBank accession number[b] | | | | Chemotype |
|---------|-----------|------|--------|------|------|------|------|--------|-----------|
| | | | | | TEF | URA | RED | Tri101 | |
| *F. vorosii* | MFG 60604 | wheat, grain | Russia, Altay Krai | 2018 | MW273184 | MW273252 | MW273218 | MW273284 | 15-AcDON |

**Notes.**

[a]NRRL–the ARS Culture Collection (USA); nucleotide sequences of these reference strains were used in phylogenetic analysis. MFG–the fungal collection of Laboratory of Mycology and Phytopathology, All-Russian Institute of Plant Protection (Russia); the studied strains.

[b]The translation elongation factor EF-1a (TEF), ammonium ligase gene (URA), reductase gene (RED), and 3-O-acetyltransferase gene (Tri101).

[c]Bold indicates the number of sequence obtained in this study.

## RESULTS

### Detection of grain infection with fungi

Visual analysis of grain samples revealed the presence of various deformities, shrunken and with a pink-white coloration of grains in the amount of 5–42% (Fig. 2). Due to prolonged wet weather, the salmon-orange conidia masses of the fungus and blue-black perithecia can be seen on the infected spikelet and glumes in barley. Most of the perithecia were mature, and when placed in a water drop, the ascospores with three septa appeared from asci.

The average germination of wheat grain was 25.1% (12–41%) and of barley grain was 55.3% (48–62%). Almost 100% infection by fungi of all grain samples was noted; often, different fungi were isolated from one grain.

Mycology analyses verified that infection by *Fusarium* spp. was the primary cause of damage in grains, and infection rates reached extremely high percentages (Table 3). Moreover, the proportion of *F. graminearum* s. lat. strains among all isolated *Fusarium* spp. averaged 83.7% in the wheat grain and 89.7% in the barley grain. *Fusarium sporotrichioides* Sherb. strains were detected in 61% of samples, but grain infection was low (1–4%) (Table S1). Among the isolated fungi, the occurrence of *F. avenaceum* (Fr.) Sacc., *F. anguioides* Sherb., *F. tricinctum* (Corda) Sacc., *F. poae* (Peck) Wollenw., *F. cerealis* (Cooke) Sacc., *F. equiseti* (Corda) Sacc., *F. incarnatum* (Desm.) Sacc., and *F. heterosporum* Nees et T. Nees as well as four strains belonging to the *Fusarium fujikuroi* species complex was lower (Table S1).

*Alternaria* spp. were the second frequent genera isolated from the grain samples. Moreover, the infection of wheat grain with *Alternaria* spp. was almost two times lower (12.9%) than that of barley grain (21.5%). *Cladosporium* spp., *Clonostachys rosea* (Link: Fr.) Schroers, Samuels, Seifert & W. Gams, *Cochliobolus* spp., *Epicoccum nigrum* Link, and other fungi were also identified in the grain mycobiota (the Table S1).

### Quantification of *Fusarium* biomass

The amount of *F. graminearum* DNA in grain flour was very high, averaging 4.9 pg/ng (Table 3). In analyzed samples of wheat grain, the amount of *F. graminearum* DNA was higher than in the barley grain samples ($p = 0.032$). The amount of 3-AcDON *F. graminearum* DNA was on average 1.3–1.1 times higher than the content of 15-AcDON genotype DNA. *F. avenaceum* DNA was detected in all grain samples in an amount that was on average 160 times less than that of *F. graminearum* DNA.

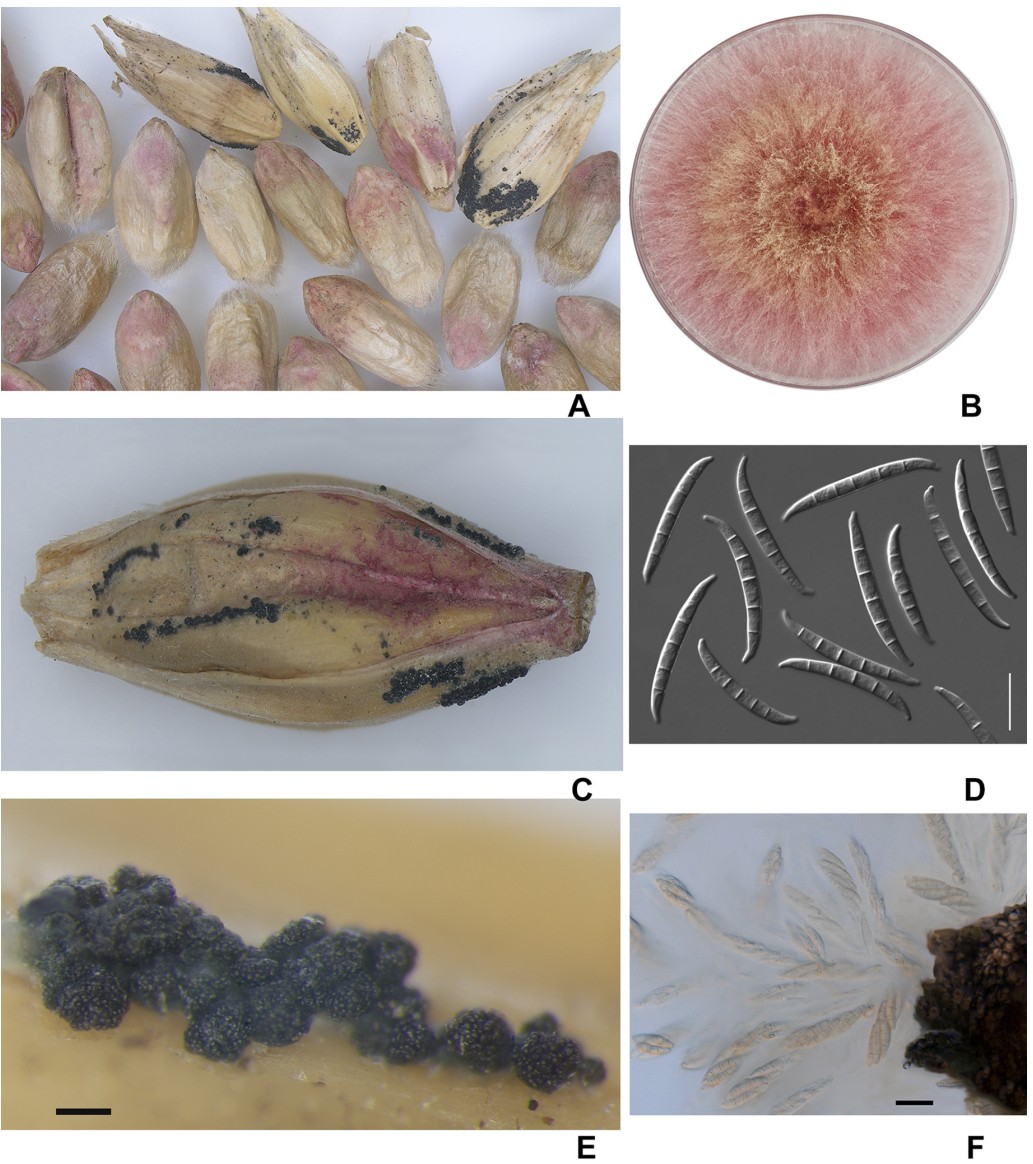

**Figure 2** The diseased wheat (A) and barley grains (C); colony of *F. graminearum* on PSA, 10 days (B); macroconidia *F. graminearum* (D); the perithecia (E); expulsion of asci and ascospores from the perithecia (F). Scale bars: D = 20 µm; E = 200 µm; F = 5 µm.

## Detection of mycotoxins

DON was found in all samples of grain. The content of DON reached 13,343 ppb in wheat samples and 7,755 ppb in barley samples. In all analyzed samples, the content of DON exceeded the maximum permissible limits (MPLs) in grain for food (700 ppb for wheat grain, 1,000 ppb for barley grain) and fodder (1,000 ppb for cereal grain), by up to 13 times (*TR TS 015/2011, 2011a*; *TR TS 021/2011, 2011b*). The exception was one barley sample, in which the DON content was lower than the MPL: 911 ppb.

**Table 3** Fungal and mycotoxin contamination of wheat and barley grain grown in the Amur region in the Russian Far East, 2019.

| Parameters | | Samples of grain | |
|---|---|---|---|
| | | **Wheat** | **Barley** |
| Grain infected with fungi on average (range), % | *Fusarium* spp. | **81.1** (58–98) | **80.5** (64–94) |
| | incl. *F. graminearum* | **68.0** (47–88) | **72.0** (61–92) |
| Content of mycotoxins on average (range), ppb | DON | **7,498** (3,207–13,343) | **5,390** (912–7,756) |
| | 3-AcDON | **122** (27–293) | **131** (0–192) |
| | 15-AcDON | **85.5** (23–179) | **93.5** (19–154) |
| | 3-DON-glucoside | **1,011** (299–2,001) | **2,128** (98–3,803) |
| | ZEN | **1,153** (92–3,670) | **537** (111–928) |
| | MON | **70.2** (10–218) | **72.7** (5–207) |
| Amount of *Fusarium* DNA $\times 10^{-3}$ on average (range), pg/ng | *F. graminearum* | **6,089** (2,658–11,342) | **2,102** (163–3,557) |
| | 3-AcDON genotype | **1,084** (395–2,007) | **508** (107–783) |
| | 15-AcDON genotype | **1,708** (755–2,776) | **371** (101–713) |
| | *F. avenaceum* | **40** (6–97) | **13** (3–38) |

In addition, other type B trichothecene mycotoxins, 3-AcDON, 15-AcDON, and DON-3gl, were detected in the grain. Of the total content of trichothecenes, the share of DON in wheat grain was 86.5% and in barley grain was 69.5%.

The content of ZEN produced by *F. graminearum* in wheat grain (92–3,670 ppb) was on average 2.1 times higher than in barley grain (111–928 ppb).

Low contents of T-2 toxin (5 and 15 ppb) and HT-2 toxin (23 and 58 ppb) produced by *F. sporotrichioides* were detected in two barley grain samples.

The MON produced by *F. avenaceum* was detected in all samples in amounts up to 218 ppb without differences between crops. The mycotoxin BEA was detected in only two wheat samples in amounts up to 13 ppb. The *fumonisins*, NEO, DAS, and fusarenone X produced by *Fusarium* fungi were not detected in the analyzed grain samples.

The mycotoxin AOH produced by *Alternaria* fungi was detected in all grain samples in small amounts (8–49 ppb). Moreover, the content of this mycotoxin in barley grain, 11.7 ppb (7.6–17.2), was significantly lower than in wheat grain, 29.0 ppb (14.2–49.1) ($p = 0.032$). AME was found in all analyzed grain samples except for two wheat samples in trace amounts. TeA was detected in all barley grains with a maximal level of 37.4 ppb and in 44% of wheat samples with a maximal level of 75.0 ppb (Table S1). Traces of TEN were found in all samples (max 6.4 ppb).

## Genotyping of *Fusarium* spp.

Multilocus analysis of the *TEF*, *URA*, *RED*, and *Tri101* sequences were used to determine the phylogenetic relationships among *Fusarium* strains. The dataset included 34 combined sequences of the analyzed strains as well as the 12 reference sequences of *Fusarium* spp. belonging to the *Fg* group and consisted of a total of 2,941 characters (612 bp from the
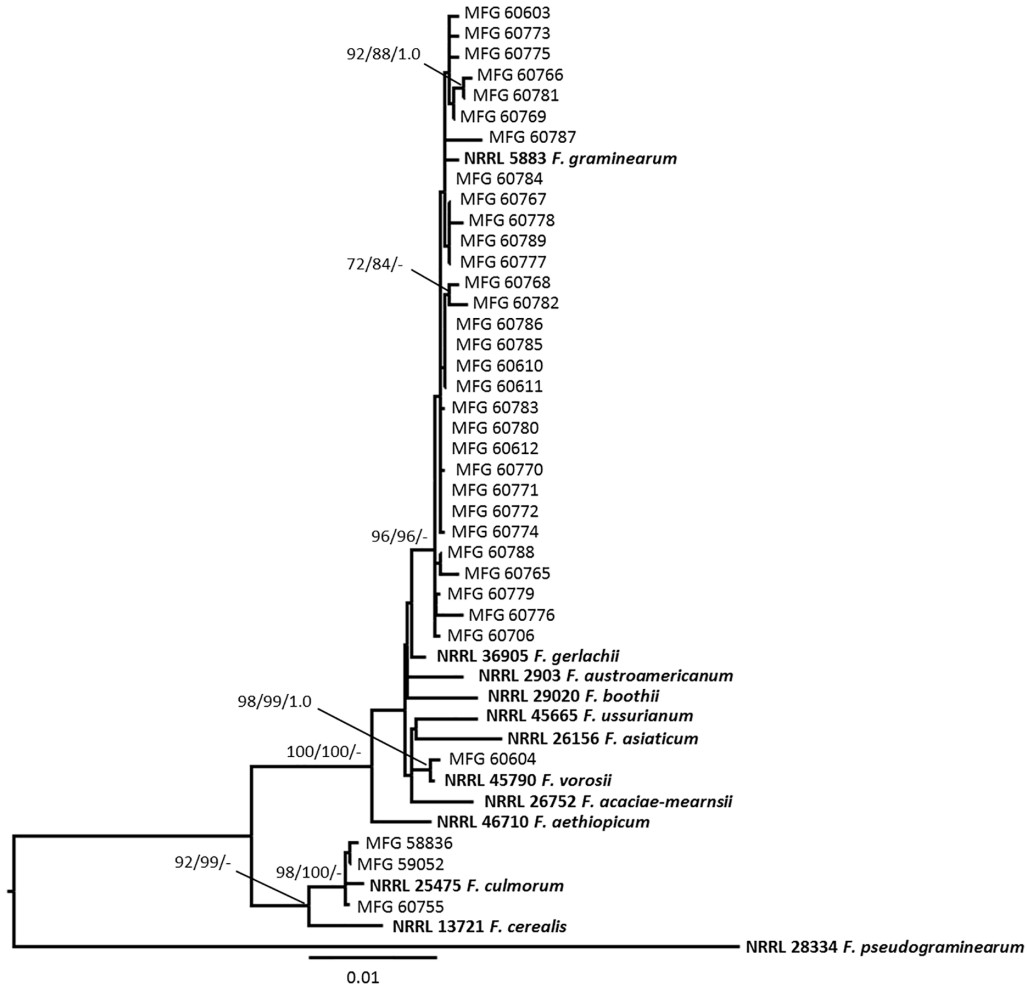

**Figure 3** **Maximum likelihood (ML) phylogenetic tree based on DNA sequence data from the fragments of translation elongation factor EF-1a (*TEF*), ammonium ligase gene (*URA*), reductase gene (*RED*), and 3-O-acetyltransferase gene (*Tri101*) of *Fusarium* species.** Numbers on the nodes are ML and maximum parsimony bootstrap support values greater than 70%, followed by Bayesian posterior probability scores greater than 0.95. Reference *Fusarium* isolates with NRRL number are indicated in bold. *F. pseudograminearum* was used as an outgroup.

*TEF*, 558 bp from *URA*, 821 bp from *RED*, and 950 bp from *Tri101*). The sequence of the *F. pseudograminearum* type strain NRRL 28334 was used as the outgroup. The resulting phylogenetic tree based on DNA sequence data of *Fusarium* species was constructed (Fig. 3). Maximum likelihood and maximum parsimony bootstrap support values greater than 70%, followed by Bayesian posterior probability scores greater than 0.95, are shown at the nodes.

The topology of phylogenetic trees constructed by different methods turned out to be similar and demonstrated the phylogenetic relationships between species established earlier (*Aoki et al., 2012*). Twenty-nine analyzed *Fusarium* strain isolated from Amur grain and one strain from Kemerovo region belonged to the clade with reference strain NRRL 5883

F. *graminearum* s. str. (Fig. 3). Among the analyzed strains of *F. graminearum* s. str., nine strains were the 3-AcDON chemotype while 21 strains turned out to be the 15-AcDON chemotype (Table 2).

From four doubtful *Fusarium* strains, one strain MFG 60604, isolated from wheat grain from the Altai Krai (West Siberia), was clustered with the reference strain *F. vorosii* NRRL 45790 with high bootstrap support (ML/MP/BP: 99/99/1.0). Our phylogenetic analysis indicates that strain MFG 60604 is *F. vorosii* and it is determined as a 15-AcDON chemotype.

Three other doubtful strains, MFG 58836, MFG 59052, and MFG 60755, formed the clade with the reference strains *F. culmorum* NRRL 25475 with high bootstrap support (ML/MP: 98/100). All three *F. culmorum* strains were the 3-AcDON chemotype (Table 2).

## DISCUSSION

Despite the long history of the FHB problem in the Russian Far East, objective data on pathogen composition and content of mycotoxins in naturally infected grain is clearly under-published. The mycological analyses of grain from this region in 2019 revealed extremely high infection of grain with *Fusarium* spp.—up to 98%. The predominant cause of FHB was the *Fg* group, which accounted for 86% of all isolated *Fusarium* spp.

Interestingly, the amount of fungal DNA in the wheat grain was on average higher than in the barley grain, while the percentage of infected grains was the same. The revealed differences may be due to the abundance of fungal biomass concentrated on the surface of barley grains (husk, palea, pericarp), while the wheat grain is completely permeated with fungal hyphae. In general, in this situation in 2019, the infection rates for both wheat and barley were off the scale. In our opinion, the highest DON content detected in this study, in the amount of 13,343 ppb, exceeds the maximum amounts of this mycotoxin in grain previously detected in the Russian territory. During the outbreak of FHB in South European part of Russia in 1985–1991 the maximal content of DON in grain reached 10,000 ppb (*Kononenko, 2005*). Recently, in 2017, DON amount of 7,920 ppb was detected in wheat grain grown in South European part of Russia (*Kononenko, Burkin & Zotova, 2020*).

The content of 3-AcDON in wheat and barley grain, as well as 15-AcDON, was similar and did not exceed 293 ppb. In the plant, DON can be present as a metabolite, DON-3gl, which is represents up to 46% of the total amount of DON in infected wheat and maize varieties (*Berthiller et al. 2009*). It has been shown that DON-3gl can be converted back to DON in mammals (*Dall'Erta et al., 2013*; *Tucker et al., 2019*). Therefore, DON-3gl is also frequently referred to as a masked mycotoxin. In our study, the maximum content of DON-3gl reached 3,803 ppb and was twice as high, on average, in barley grain than in wheat grain. The amounts of DON-3gl come to 13.5% and 39.5% of the total amounts of DON in infected wheat and barley samples, respectively. However, there were no significant differences in the content of the trichothecene mycotoxin average between wheat and barley grains.

Using morphology to accurately assess species limits for the *Fg* group is not reliable. Before this study, we hypothesized that in the extremely humid and warmest conditions

of 2019, in the area where FHB outbreaks were observed for at least 140 years, several species of the Asian clade of the *Fg* group will be identified. Especially considering that earlier we have already found three species of the *Fg* group in this region: *F. graminearum* s. str., *F. ussurianum* and *F. vorosii* (*Yli-Mattila et al., 2009*). Selecting freshly isolated fungi for analysis, we took cultures for a detailed study, which included all the morphological diversity present within the limits possible for the *Fg* group (pigmentation, rate of formation of macroconidia, size, and shape). Multilocus phylogenetic analysis revealed that all strains from the Amur grains belonged to the *F. graminearum* s. str.

Molecular methods make it possible to reveal the intraspecific diversity of *F. graminearum* and to establish the quantitative presence of two different chemotypes. The *F. graminearum* strains are divided into 3-AcDON and 15-AcDON chemotypes depending on the prevailing acetylated form of DON (*Alexander et al., 2011*; *Foroud et al., 2019*). Regional differences have been reported regarding the occurrence of chemotypes within the *Fg* group (*Foroud et al., 2019*; *Pasquali et al., 2016*). In our study, on average, the DNA content of the 3-AcDON and 15-AcDON chemotypes of *F. graminearum* in the grain was similar, but the DNA of the 15-AcDON chemotype in wheat grain was significantly higher (4.6 times) than in barley ($p = 0.014$), whereas the difference in DNA content of the 3-AcDON fungus chemotype in wheat and barley grain was insignificant. It is not known whether the observed differences are related to chemotype-specific plant-host preferences. There may be a difference in pathogenicity between the 3- and 15-AcDON chemotypes to wheat and barley (*Foroud et al., 2019*; *Clear et al., 2013*).

According to our results, 30% of the analyzed *F. graminearum* strains were the 3-AcDON chemotype, while 70% of the strains were the 15-AcDON chemotype. Previously, the chemotype analysis of the 105 *F. graminearum* strains collected in the Russian Far East in 1998–2006 revealed the approximately equal occurrence of 3-AcDON (48%) and 15-AcDON (52%) chemotypes (*Yli-Mattila et al., 2009*). An increase in the 15-AcDON chemotype has recently been shown in regions of Europe, where the 3-AcDON chemotype was previously dominant, although many of the factors affecting their distribution are still unclear (*Nielsen et al., 2012*; *Aamot et al., 2015*; *Pasquali et al., 2016*; *Foroud et al., 2019*). The third chemotype of *F. graminearum* s. str. producing nivalenol (NIV) has not yet been identified in Russia or China (*Shen et al., 2012*), although it is known to be found in Europe (*Pasquali et al., 2016*).

In our analysis, *Fusarium* sp. strain MFG 60604 was included that was isolated from wheat grain in the West Siberian region (the Altai Krai); phenotypically, this strain was a dubious representative of the *Fg* group. In this region, the occurrence of *F. graminearum* was previously not typical, but in recent years, we have been identifying this pathogen in cereal grains (*Gagkaeva et al., 2019*). The strain MFG 60604, isolated from wheat grain from the West Siberia, was clustered with the reference strain *F. vorosii* NRRL 45790 with high bootstrap support (ML/MP/BP: 98/99/1.0), which allows for accurate establishment of its species affiliation. A single strain (MFG 60604) identified as *F. vorosii* in this study, is the only third strain of *F. vorosii* found in Russia and the first one identified in the Siberian region. Previously identified strains of *F. vorosii* from the Russian Far East belonged to the 15-AcDON chemotype (*Yli-Mattila et al., 2009*) and so did the strain identified in

this study. However, among six *F. vorosii* strains originating from Korea, five were the NIV chemotype, while only one was the 15-AcDON (*Lee et al., 2016*). Among *F. vorosii*, no strains of the 3-AcDON chemotype have been identified, which, probably, were not detected due to the small number of strains of this species analyzed at present. In the limited surveys at present, strains of several species of the *Fg* group were found to represent only a single chemotype (*Aoki et al., 2012*).

Two strains of *F. culmorum* of closely related taxon to *Fg* group from the West Siberian and Ural regions and one from the South European region of Russia were included in the study. The high genetic similarity of analyzed *F. culmorum* strains collected from remote regions characterized by different climatic conditions (the distance between isolation points is about 2,500 km) is consistent with the previously shown information that *F. culmorum* is a single phylogenetic species with little or no differences between lineages, despite the geographic separation of genotypes (*Obanor et al., 2010*).

The studies analyzing the occurrence of *F. culmorum* chemotypes in different regions, as a rule, show a significant excess of the occurrence of the DON chemotype compared to the NIV chemotype (*Laraba et al., 2017*; *Pasquali et al., 2016*; *Scherm et al., 2012*). Strains of the 15-AcDON chemotype typical for *F. graminearum* were not identified among the strains of *F. culmorum*. A previous analysis of a few strains of *F. culmorum* from the Russian territory has also characterized them as the 3-AcDON chemotype (*Yli-Mattila et al., 2009*).

*Fusarium* spp. continue to pose a threat to farmers, destroying crops or dramatically reducing yields, as well as to animal and human health due to the production of mycotoxins. Even in our time, when we know much more about the nature of *Fusarium* spp. then 140 years ago, we are still not very successful in controlling the diseases they cause on crops if conditions are favorable for the development of pathogens. Indeed, in the process of our study, it was shown in the mass media that although the grain was harvested with great difficulty, due to the significant contamination of the grain, part of the crop, 240 tons, had to be destroyed by fire.

## CONCLUSIONS

The high prevalence of Fusarium head blight in cereal grains cultivated in the Far East is particularly alarming and strongly indicates the need for increased measures to prevent plant infection and improved food safety interventions. The maximum DON content in wheat grains reached 13,141 ppb in this study. The multilocus sequence revealed that the majority of the strains used in this study belonged to *F. graminearum* s. str.

## ACKNOWLEDGEMENTS

We are grateful to the managers of the Russian Branch of BASF company for their help with the grain sample collection and Nadezhda N. Gogina from the All-Russian Scientific and Technological Institute of Poultry (Moscow region) for carrying out HPLC-MS/MS analysis.

### Funding

This work was supported by the Russian Science Foundation (project no. 19-76-30005). The funders had no role in study design, data collection and analysis, decision to publish, or preparation of the manuscript.

### Grant Disclosures

The following grant information was disclosed by the authors:
The Russian Science Foundation (project no. 19-76-30005).

### Competing Interests

The authors declare there are no competing interests.

### Author Contributions

- Tatiana Gagkaeva, Aleksandra Orina and Olga Gavrilova conceived and designed the experiments, performed the experiments, analyzed the data, prepared figures and/or tables, authored or reviewed drafts of the paper, and approved the final draft.

### Field Study Permissions

The following information was supplied relating to field study approvals (i.e., approving body and any reference numbers):
The collection of specimens at a fields was approved by the Russian Science Foundation (project number: 19-76-30005).

### DNA Deposition

The following information was supplied regarding the deposition of DNA sequences:
The DNA sequences are available at GenBank: MW273157 to MW273258.

### Data Availability

Raw data are available in the Supplemental Files.

### Supplemental Information

Supplemental information for this article can be found online at http://dx.doi.org/10.7717/peerj.12346#supplemental-information.

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
