# Peer review of "Fusarium head blight in the Russian Far East: 140 years after description of the ‘drunken bread’ problem"

_PeerJ, doi:10.7717/peerj.12346_

## Round 0.1 · original submission · Major Revisions

Dear Dr. Gagkaeva,

Your manuscript had reviewed by three experts in your research areas. Two reviewers have recommended minor revisions but the third reviewer has rejected your manuscript in the current form. Based on my assessment, I invite you to MAJOR revision and resubmission.

I think the title of the manuscript “Fusarium head blight in the Russian Far East: 140 years of the 'drunken bread' problems” sounds like “a review paper’ and seems to be a very broad title. Please change it with a short and clear title in a line of your research.

The third reviewer pointed out several shortcomings in your manuscript. These included a lack of clear objectives, hypotheses, proper experimental design, and sampling strategies, and location descriptions.

The identification of the Fusarium species complex is based on morphology. Considering the diversity and species complex, a robust molecular approach is necessary to identify Fusarium species.

For mycotoxin analysis, whether DNA was extracted from grain or the mycelia. Please ratify. Sequences are not yet available on GenBank.

The authors need to submit sequence data of each isolate and provide accession numbers.

All figure qualities need to improve and tables must be self-explanatory. The Introduction, results, and discussion must be concise and precise.

Thorough English editing is necessary for clarity.

The second reviewer’s comments (pdf file) are attached.

Once I received your revised manuscript, I will forward it to the same reviewers or find new reviewers for further review. Therefore, I ask you to carefully read all comments provided by the reviewers and revise your manuscript.

·

Basic reporting

The article is clearly written, intelligible, and provides a coherent narrative. GenBank accession numbers are provided; authors' sequences are not yet available on GenBank but that is common practice (to not release sequences until publication) and the authors provided a FASTA file with the sequences and other raw data in supplementary files.

Experimental design

The paper reports original primary research which falls within the Aims and Scope of PeerJ. The detailed analysis of species and mycotoxin composition of organisms causing Fusarium head blight of wheat and barley in the Russian Far East is a valuable contribution to the growing body of literature detailing Fusarium species distribution, especially as those distributions are changing in many parts of the world. Methods used are standard in the field, and are described and/or referenced such that they can be repeated.

Validity of the findings

Findings and conclusions are justified by the data and the methods.

Additional comments

Here are a handful of minor edits and suggestions:
Lines 27, 245: Values like “6089 x 10^3” and “2102 x 10^3” are not standard; more commonly I would expect to see “6.1” or “6.1 x 10^0”
Line 88: suggest “endemic Asian species”
Line 128: probably not necessary to specify “self-made”
Line 148: suggest “normalized” for “aligned”
Line 179: “SecurityGuard” (no space; capital “S” and “G”); this is a trade name for the brand of cartridge. “Security guard” (with a space; lowercase) has a different meaning.
Line 230: “verified that infection by Fusarium fungi was the primary cause” or “verified that Fusarium fungi was the primary causal agent”
Line 241: suggest leaving out “The representatives of” and starting the sentence with “Cladosporium spp.”
Lines 253-256: rephrase; suggest “exceeded the maximum permissible limits (MPLs) in grain for food (700 ppb for wheat grain, 1,000 ppb for barley grain) and for fodder (1,000 ppb for cereal grain) (TR TS 015/2011; TR TS 021/2011), by up the 13 times.” As written, it sounds as though all samples had 13x as much DON as MPLs, while what you want to say is that all samples exceeded MPLs, and the highest levels were up to 13 times MPL.
Line 327: suggest “Using morphology” instead of “The morphological recognition”
Figure 1: “were drawn”
Figure 2: suggest “expulsion of asci and ascospores”

Reviewer 2 ·

Basic reporting

No comments

Experimental design

No comments

Validity of the findings

No comments

Additional comments

At some places language needed to be improved for clarity and in some others the run on sentences needs to be broken down into shorter sentences for better clarity to the readers.

Annotated reviews are not available for download in order to protect the identity of reviewers who chose to remain anonymous.

Reviewer 3 ·

Basic reporting

The author tried to explain very well regarding writing, but the message delivery was not precise; for example, the research objective was not clear, the writing is not in flow, and information was scattered all around. The material and methods and result section should be clear and consistent.

This article needs to rewrite with a clear objective, hypothesis, methods, result, and discussion.

Experimental design

I think the experimental design was not enough to support for aims and scope of the journal. No clear objective or hypothesis was given. Sampling methods, sample location, or distribution pattern were not clear. The morphology is not enough to identify the Fusarium species complex. Only 29 samples collected in a single season are not enough to represent the whole story of the genetic relationship in this complex pathosystem.

Validity of the findings

Finding is valid for the amount of data collected. However, it is not clear how sampling was done, how isolates were selected, and which isolate was used for which purpose. For example, grain/isolate used for mycotoxin analysis, the sample used in DNA extraction from grain or the mycelia, selection criteria of 29 isolates, etc.

Additional comments

Dear Authors,
I enjoyed reading your manuscript and understand the problem of Fusarium Head blight in Far East Russia for such a long time. You have collected nine spring wheat and four barley samples from the Amur region of Russia (Far-east) in August 2019. Samples used for mycotoxin analysis, isolate collection, or species characterization based on morphology. qPCR was done to quantify F. graminearum and F. avenaceum or 3-AcDON and 15-AcDON chemotypes of F. graminearum. You have also performed a genetic analysis of 29 isolates using TEF, URA, RED, and Tri101 gene sequences.

However, this article lacks a clear objective or hypothesis; the abstract is too long, the sampling method is not clear, the identification of species based on morphology is not enough, the number of isolates used in the genetic analysis is insufficient, and data based on a single season lacks actual representation. I think this article doesn't meet the requirements to be published in PeerJ with such flaws.
Thanks

---

## Round 0.2 · Major Revisions

Dear Dr. Gagkaeva,

I do think that the work is novel and interesting enough due to the same sample sizes analyzed.; however, it must be clearer. PeerJ aims to set the highest standards in the field for publication. Based on the comments of the reviewer(s) and my own assessment, your manuscript will require MAJOR revisions prior to further consideration for publication. More importantly, please change the title, if possible. The abstract is confusing and should be re-written with clear hypotheses. Several important points are missing (e. g. sampling strategies, sample sizes, location, varieties name from which these fungi were collected) in the Materials and Methods. Interpretation of the Results (pathogenicity test, if any, and molecular analysis) needs to describe precisely. The whole section of the Discussion is needed to restructure to better explain the findings of the results.

Please make sure methods and experimental set-ups are clear, data and results are well described and data are deposited on appropriate websites. The discussion should be robust and comprehensive. The implication of this study should be written at the end of the Discussion. The manuscript needs extensive English editing by an English-speaking editor.

I have some MAJOR concerns about the paper, which is attached (pdf file). These things, I would like to see changed in the revised draft.

Please revise your manuscript using a word processing program and save it on your computer. Please also highlight the changes to your manuscript within the document by using the track changes mode in MS Word or by using bold or colored text.

I am also reserving the right to further send your revised manuscript out to the original reviewer(s) or to the new reviewer(s) before making a final decision on suitability for publication.

Thank you for submitting your paper to PeerJ.


Best regards,

Sincerely,

Tika Adhikari

·

Basic reporting

English is not that of a native speaker but is clearly understood.

Experimental design

No comment

Validity of the findings

No comment

Additional comments

The authors have addressed the concerns I raised in the previous review. Further, I agree with their argument for retaining their proposed title, and they have bolstered their identificatiosn with appropriate DAN studies (EF1A, in particular) for identifying morphologically-indistinguishable members of the Fusarium graminearum species complex, and/or used "Fusarium graminearum sensu lato" when discussing taxa idenitfied based on morphology.

---

## Round 0.3 · Minor Revisions

The authors,

There are still several misspellings and grammatical errors. For clarity, extensive English editing is required before publication. Please use the attached MS Word file for further editing.

Thank you.

---

## Round 0.4 · accepted · Accept

Dear Dr. Gagkaeva,

Your manuscript has been Accepted for publication.

Congratulations.


Tika Adhikari